# Effects of Antimicrobial Photodynamic Therapy on Organic Solution and Root Surface In Vitro

**DOI:** 10.3390/antibiotics10020101

**Published:** 2021-01-21

**Authors:** Yuji Matsushima, Akihiro Yashima, Meri Fukaya, Satoshi Shirakawa, Tomoko Ohshima, Tomomi Kawai, Takatoshi Nagano, Kazuhiro Gomi

**Affiliations:** 1Department of Periodontology, Tsurumi University School of Dental Medicine, 2-1-3 Tsurumi, Tsurumi ku, Yokohama 230-8501, Japan; matsushima-y@tsurumi-u.ac.jp (Y.M.); yashima-akihiro@tsurumi-u.ac.jp (A.Y.); fukaya-meri@tsurumi-u.ac.jp (M.F.); shirakawa-satoshi@tsurumi-u.ac.jp (S.S.); nagano-takatoshi@tsurumi-u.ac.jp (T.N.); 2Department of Oral Microbiology, Tsurumi University School of Dental Medicine, 2-1-3 Tsurumi, Tsurumi ku, Yokohama 230-8501, Japan; ohshima-t@fs.tsurumi-u.ac.jp (T.O.); kawai-tomomi@tsurumi-u.ac.jp (T.K.)

**Keywords:** photodynamic therapy, antibacterial therapy, bacterial flora, dental treatment, lipopolysaccharide, organic resolution

## Abstract

Antimicrobial photodynamic therapy (a-PDT) is attracting attention as a new form of dental treatment. While it is primarily applied to produce an antibacterial effect, it decreases lipopolysaccharide (LPS) and protease activity. Here, we evaluated differences in the antibacterial activity of a-PDT on three types of bacteria and the effects on the organic substances (i.e., albumin and LPS). Furthermore, we investigated the effects of a-PDT on root surfaces. A FotoSan630^®^ and toluidine blue were used to perform a-PDT in this study. We measured its antimicrobial activity against *Porphyromonas gingivalis*, *Streptococcus mutans*, and *Enterococcus faecalis*. Antimicrobial testing revealed strong antimicrobial action and *P. gingivalis*, *E. faecalis*, and *S. mutans* were almost undetectable after 50, 120, and 100 s, respectively. In organic resolution tests, albumin was significantly decreased from 1 min after a-PDT application onward, while LPS significantly decreased at 5 min after the application. The root surfaces after a-PDT were confirmed to be cleaner than the controls without suffering any damage. Depending on the bacterial species, a-PDT exhibited antimicrobial activity against various types of bacteria and sensitivity differed. Moreover, we reported that a-PDT resolves protein and LPS, enabling the formation of a healthy root surface without any damage.

## 1. Introduction

Recently, photodynamic therapy (PDT) has been clinically applied as a dental treatment [1,2]. PDT that targets bacteria in particular is called antimicrobial photodynamic therapy (a-PDT). These therapies are beginning to be used to eliminate bacterial infections in periodontal disease [3,4,5,6], infected root canal treatment [7,8,9,10], and caries treatment [11,12,13,14]. There are three elements required for a-PDT: a photosensitizer, an excitation light, and oxygen. When a light-sensitive agent is exposed to light of an excitation wavelength, it becomes a triplet state as per the intensity of light. When an excited triplet state photosensitizer directly reacts with a biological substrate, it produces free radicals, which react with bio-oxygen molecules to produce various reactive oxygen species (ROS). Furthermore, singlet oxygen is produced when it reacts directly with oxygen molecules [15,16]. The primary pathway is considered to be the oxidation of proteins, nucleic acids, and lipids of living organisms or bacteria by the production of these ROS, particularly singlet oxygen, which damages the cell membrane and cell walls of bacteria and thereby obtains a bactericidal effect [17,18,19,20]. In particular, in periodontal disease, bacteria and bacterial-derived internal toxins (lipopolysaccharides; LPS) induce inflammation in the periodontal tissue, causing the destruction of the periodontal tissue and worsening of the pathological condition. The basic treatment for periodontal disease is the mechanical removal of bacteria and inflammatory substances, such as scaling and root planing (SRP). However, removing plaque bacteria or LPS that are adhered to the deep periodontal pocket in which periodontal disease has progressed or to the root surface with complicated morphology using a mechanical method such as SRP is difficult. It is complicated to manipulate the device to reach the root surface. This difficulty is attributed to the morphology. Furthermore, local or systemic administration of antibacterial agents is performed, but there are potential problems with this approach, such as causing the development of resistant bacteria. Because a-PDT does not have the problem of the emergence of resistant bacteria, multiple studies have been performed with the aim of improving the therapeutic effect, using it in combination with treatments such as SRP. In reports of the combined use of a-PDT with SRP for chronic periodontitis, many studies have suggested that there was significant improvement in clinical parameters compared to using SRP alone [21,22,23,24,25,26,27,28,29,30]. Nevertheless, there was no statistical difference in the clinical parameters between the group treated with SRP under systemic antibiotics and the group treated with a-PDT and SRP [3] or patients with periodontitis. There was no statistical difference in probing depth (PD) and clinical attachment level (CAL) when conventional periodontal therapy (SRP) and a-PDT were compared in patients with peri-implantitis [5]. In addition to these studies, the effects of a-PDT on antibacterial and inflammatory cytokines have been investigated. Dobson et al. showed that a-PDT significantly reduced the number of periodontal pathogens [31], and Kömerik et al. revealed that they reduced LPS and protease activity in Gram-negative bacteria (*Escherichia coli* 055-B5) [32]. Thus, it is considered that a-PDT can be expected to have bactericidal action and to inactivate harmful substances derived from bacteria attached to the root surface. This has the potential to give dental treatment an edge. However, few studies have determined whether a-PDT directly reduces or inactivates inflammatory substances such as LPS.

In this study, we morphologically evaluated the effect of a-PDT on the root surface using a phase-difference electron microscope. Furthermore, we investigated whether a-PDT reduces organic substances such as albumin, which is a common protein, or LPS, which is an endotoxin derived from periodontopathic bacteria. Moreover, we evaluated the antibacterial activity of a-PDT against periodontopathic bacteria (*Porphyromonas gingivalis*), caries pathogenic bacteria (*Streptococcus mutans*), and refractory root canal bacteria (*Enterococcus faecalis*). If the effect of a-PDT is not only antibacterial but also results in the decomposition of organic substances such as LPS, it is considered that the applicable range of a-PDT will be further expanded.

## 2. Results

### 2.1. Effects on Root Surfaces

Scanning electron microscopy (SEM) observation confirmed a smeared layer and sediment on the root surfaces in the control group that only underwent SRP (Figure 1a). In the a-PDT group, comparisons revealed that, although the smeared layer remained, there was less sediment attached to the root surfaces (Figure 1b).

### 2.2. Organic Resolution

Investigation with electrophoresis revealed a decreased bandwidth for albumin in an application time-dependent manner compared with the control group. While the bandwidth decreased with longer application times, it did not completely disappear (Figure 2). A BCA (bicinchoninic acid) protein assay was used to evaluate the albumin volume in specimens under the same conditions as for electrophoresis. The blue pigment of the toluidine blue in the specimens was able to be completely removed with a spin gel filtration column, and absorbency measurement was not affected. However, gel filtration caused the albumin volume to decrease by ~30%. In the specimens that underwent a-PDT for 1, 2, and 5 min, absorbency significantly decreased, coinciding with the results from electrophoresis. Although albumin resolution was able to be confirmed, it was not completely resolved (Figure 3).

### 2.3. LPS Resolution

The toluidine blue was removed with a spin gel filtration column, and the LPS in the specimens was measured. No decreases in LPS were noted for applications of 1 or 2 min. However, a significant decrease was noted in LPS for an application of 5 min. Gel filtration did not result in a decrease in LPS (Figure 4).

### 2.4. Antimicrobial Action

The results of bacterial culture testing indicated that bacterial counts of *S. mutans*, *E. faecalis*, and *P. gingivalis* were decreased in the application group compared to the control group in a time-dependent manner (Figure 5). The measurement of bacterial counts revealed that the time at which the bacterial counts started to decrease differed depending on the bacterial species. *P. gingivalis* was significantly decreased at 30 s and was undetectable at 50 s. *E. faecalis* was decreased significantly at 90 s and was undetectable at 120 s. *S. mutans* was decreased significantly at 80 s and was undetectable at 100 s, indicating strong antimicrobial action against each of these bacterial species (Figure 6a–c). Thus, while differences were observed depending on the bacterial species, strong antimicrobial effects were confirmed overall.

## 3. Discussion

Previously, studies reported that using a-PDT with SRP for dental treatment improves the clinical parameters for periodontal disease [21,22,23,24,25,26,27,28,29,30]. However, few studies have investigated the effects of a-PDT on the root surface and on organic substances. Many unclear aspects remain regarding differences in antimicrobial action depending on the bacterial species. Therefore, in this study, we used a Fotosan630^®^ (CMS Dental, Copenhagen, Denmark) combining 620–640-nm red LED light and toluidine blue to perform a-PDT and then conducted morphological observation using SEM to determine how a-PDT affects the root surface. The results seemed to indicate that the a-PDT group exhibited cleaner root surfaces, compared to the control group that only underwent SRP. It was considered that the reason was that the root surface was cleaned by decomposing the organic matter on the root surface. Therefore, we investigated its organic resolution with albumin. The results indicated that the albumin amount decreased from after one minute of a-PDT application (Figure 2). The quantitative measurement of the amount of albumin resolved indicated that it significantly decreased compared to the control with an application of one minute and longer. Note that additional decreases were observed with five minutes of application. This demonstrated that a-PDT exhibits organic resolution. This action may be attributed to the oxidization caused by the production of ROS and, in particular, singlet oxygen induced by a-PDT [17] (Figure 3). Next, we used a chromogenic LAL (limulus amebocyte lysate)endotoxin assay to investigate the effects of a-PDT on LPS, an endotoxin that is a cell wall component of Gram-negative bacteria. The results demonstrated that while there was no significant decrease in LPS with up to two minutes of application, it was decreased significantly with five minutes of application. Kömerik et al. [32] previously reported that photodynamic action decreased LPS and protease. In this study, although we noted LPS resolution, it took more time until the resolution occurred compared to albumin (Figure 4). This may have been because lipid A of LPS has a chemical structure that involves multiple fatty acid chain bonds, thus giving it stronger carbon bonds than albumin, which has peptide bonds. We used bacteria that cause periodontal disease (*P. gingivalis*), bacteria that cause dental caries (*S. mutans* strain), and bacteria that cause refractory apical periodontitis (*E. faecalis* strain) to investigate the antimicrobial activity of a-PDT. The results indicated that a-PDT exhibits antimicrobial action against all three bacterial species. The bacterial count of *P. gingivalis*, a Gram-negative bacterium, started to decrease significantly in the 30 s application group compared to the control group and was almost undetectable after 50 s of application. However, *E. faecalis* and *S. mutans*, both Gram-positive bacteria, decreased significantly compared to the control group at 90 and 80 s, respectively, and were undetectable at 120 and 100 s, respectively. The longer application time required than for *P. gingivalis* may have been attributed to differences between Gram-negative and Gram-positive bacteria in terms of cell wall thickness, cell membrane structure, or because of differences in the permeability of the photosensitive agent into the bacteria [17]. However, strong antimicrobial action against all three bacterial species was confirmed. This experiment demonstrated that a-PDT exhibits organic resolution capabilities, suggesting that a-PDT could be applied to the root surface to clean it and remove any organic debris. Moreover, it could be applied to various fields because it exhibited antimicrobial effects against not only bacteria that cause periodontal disease but also bacteria related to caries and endodontic lesions. The resolution of LPS, which is an endotoxin, was confirmed. This suggested that the irradiation of a-PDT to sites that are difficult to approach mechanically, such as the furcation area or deep periodontal pockets, could enable the removal of LPS. Moreover, as a-PDT can act non-invasively inside periodontal pockets and, unlike antibiotics, does not cause the emergence of resistant bacteria, it appears to be an excellent auxiliary treatment technique for chronic periodontitis. However, it is associated with various problems, such as relatively long light irradiation times being required to enable sufficient antimicrobial and organic resolution capabilities to be exhibited. To overcome this issue, photosensitive agents that produce additional free radicals, wavelengths that can more efficiently excite photosensitive agents, and powerful irradiation devices need to be developed going forward. Our results demonstrated that a-PDT exhibits antimicrobial activity against various types of bacteria. Moreover, it exhibited organic resolution capabilities and showed the potential to remove organic debris and LPS that adhere to the root surface and clean the root surface without damaging the root surface.

## 4. Materials and Methods

### 4.1. Effects on Root Surfaces

To evaluate the effects on the root surface after a-PDT was applied, we used a Gracey Curette #5/6 (Hu-Friedy, Chicago, Illinois, USA) on extracted human teeth (maxillary anterior teeth and mandibular premolars with uncomplicated root morphology were used) that had undergone 10 strokes of root planing. The control group comprised teeth of which the root surface only underwent rinsing with water for 10 s. The root surfaces of teeth in the a-PDT group were coated with 0.06 mL of toluidine blue (0.01%; FotoSan Agent: Toluidine Blue O, 0.01%, CMS Dental, Copenhagen, Denmark), had FotoSan630^®^ light applied for 90 s, and were then rinsed with water for 10 s. After the treatment, the critical point was dried, gold was vapor-deposited, and the tooth root surface was visually observed by the same researcher at a 3000-fold magnification using a scanning electron microscope (S-4800, Hitachi High-Technologies Corporation, Tokyo, Japan).

### 4.2. Organic Substances Resolution

Albumin solutions were mixed such that the final concentration of albumin (Bovine Serum, Wako Pure Chemical Corporation, Osaka, Japan) was 2 mg/mL and the final concentration of toluidine blue O (Chroma Gesellschaft Schmidt & Co, Münster, Germany) was 0.01%. Next, the FotoSan630^®^ was used to apply light from the base of transparent test tubes for 0, 1, 2, and 5 min in the following experiments.

### 4.3. Electrophoresis Experiment

The recovered specimens underwent electrophoresis under conditions in accordance with the method described by Laemmli [33] (40 mA for 80 min with a 15% polyacrylamide gel containing 1% SDS). They were then dyed with Coomassie brilliant blue (Ezstain Aqua, ATTA, Tokyo, Japan) and depigmented with distilled water, after which bands were detected. Protein MultiColor III (Range 17–213 kDa) (Bio Dynamics Laboratory Inc, Tokyo, Japan), a standard marker, was used to determine the molecular weight.

### 4.4. Albumin Volume Measurement

Protein measurement was performed with the Pierce BCA Protein Assay Kit (Thermo Scientific, Rockford, IL, USA) [34] to quantify the volume of albumin in specimens. As toluidine blue affects protein detection in specimens, analysis was performed after removing it with a spin gel filtration column (Centri-Sep; Princeton Separations, Inc., Freehold, NJ, USA) and Sephadex G-50 (molecular cutoff range: 1.510 kDa) (Centri-Sep; Princeton Separations, Inc., Freehold, NJ, USA). [35] We then transferred 100 µL of each of the specimens into a 96-well multi-plate and measured absorbency with a wavelength of 562 nm.

### 4.5. LPS Resolution

To investigate the effects of a-PDT on LPS, we adjusted the specimens by adding toluidine blue to the LPS solution such that the final concentration of LPS was 1 EU/mL and the final concentration of toluidine blue was 0.01%. After applying light with a Fotoso630^®^ from the test tube base for 0, 1, 2, and 5 min, the toluidine blue was removed using a spin gel filtration column. LPS detection was performed in accordance with the Toxin Sensor Endotoxin Detection System (GenScript USA Inc, Piscataway, NJ, USA) manual. We then transferred 100 µL of each of the specimens into a 96-well multi-plate and measured absorbency with a wavelength of 545 nm.

### 4.6. Measurement of Antimicrobial Action

To evaluate the antimicrobial action of a-PDT, we used bacteria that cause periodontal disease (*P. gingivalis:* strain ATCC33277), bacteria that cause dental caries (*S. mutans* strain: ATCC2517), and bacteria that cause refractory apical periodontitis (*E. faecalis* strain: ATCC19433). The bacterial counts were 1 × 10^6^ CFU (colony forming unit)/mL. The amount of bacteria was adjusted using a spectrophotometer (UV-1200, Shimadzu Corporation, Kyoto, Japan) with absorbency at 620 nm. Toluidine blue was added to the 100 μL bacterial solution such that the final concentration was 0.01%, and the solution was then mixed. Light was applied with a Fotoso630^®^ for 0, 30, 40, 50, 60, 80, 90, 100, 110, 120, and 300 s. The bacterial solutions collected after light applications were completed were then used to seed the agar medium. The *S. mutans* was placed in a TS (trypticase soy) agar medium and the *E. faecalis* was placed in a BHI (brain heart infusion) agar medium and they were cultured at 37 °C in a thermostatic incubator. *P. gingivalis*, which is an anaerobic bacteria, underwent anaerobic culturing with 5% Merino sheep defibrinated blood (Nippon Bio-Test Laboratories Inc., Saitama, Japan) and 5 µL/mL hemin and menadione (Wako Pure Chemical Corporation, Osaka, Japan) added to a Brucella HK agar medium using the anaerobic culture equipment Anaerobox ANX-1 (Hirasawa Corporation, Tokyo, Japan) with nitrogen at 80%, hydrogen at 10%, and carbon dioxide at 10% at 37 °C for seven days. After culturing, we counted the number of colonies in each of the media.

### 4.7. Statistical Analysis

A one-way ANOVA followed by Tukey’s test was used to perform multiple comparisons of the differences over time between organic and LPS resolutions. The level of statistical significance was set as *p* < 0.05. For the bacterial culture test, the Mann–Whitney U test was used and *p* < 0.01 was considered to be significantly different. All statistical analyses were performed with SPSS Statistics version 19 (IBM, Tokyo, Japan). All experiments were performed in duplicate and repeated at least three times in independent experiments.

## 5. Conclusions

The a-PDT exhibits antimicrobial activity against various types of bacteria, although some slight differences are depending on the bacterial species. Moreover, a-PDT exhibits organic resolution capabilities, including LPS, and it was shown the possibility that a-PDT could be used to form bacteriologically and chemically clean root surface without damaging the root surface.

## Figures and Tables

**Figure 1 antibiotics-10-00101-f001:**
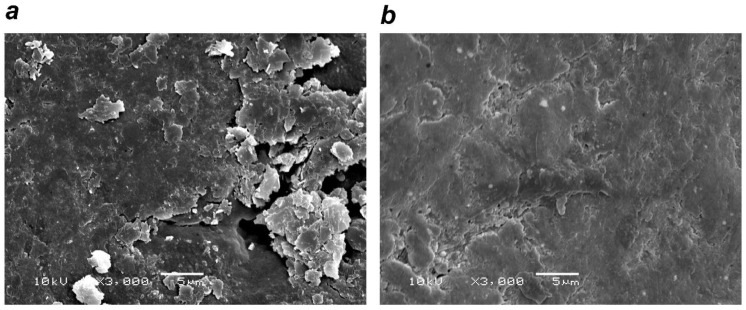
Effects of (antimicrobial photodynamic therapy) a-PDT on the root surface. (**a**) Control group (rinsing after scaling and root planing (SRP)): smeared layer and sediment noted on the root surface. (**b**) a-PDT group: decreased sediment on the root surface. (Scanning Electron Microscope image (SEM) × 3000)

**Figure 2 antibiotics-10-00101-f002:**
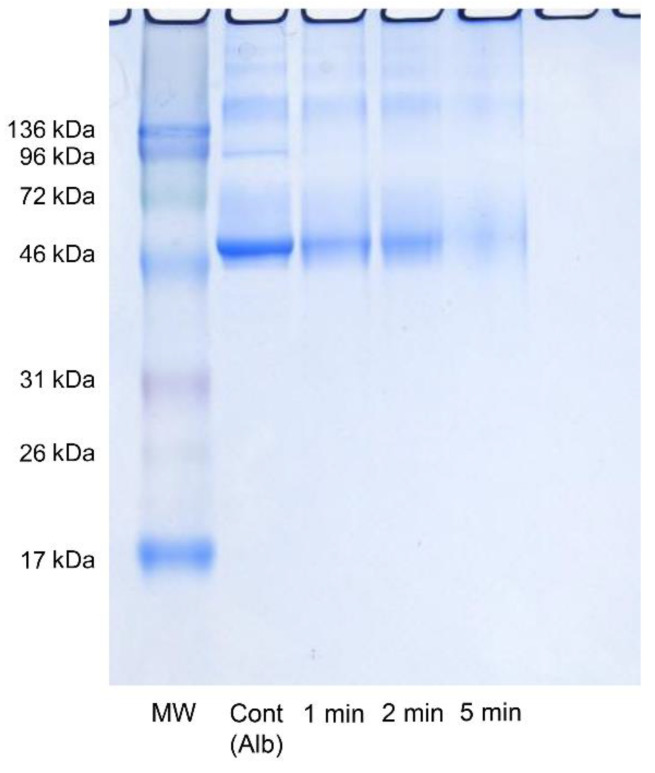
Effects of a-PDT on albumin (electrophoresis image). Light was applied with FotoSan630^®^ to the albumin suspension for 0, 1, 2, and 5 min. The albumin bandwidth decreased in an irradiation time-dependent manner. MW: molecular weight marker.

**Figure 3 antibiotics-10-00101-f003:**
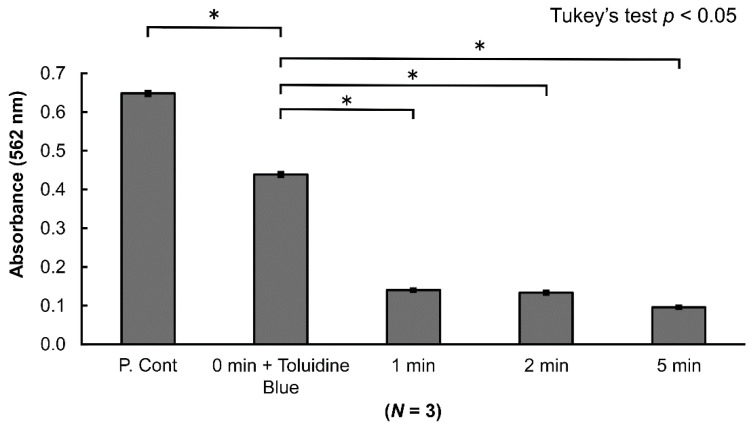
Effects of a-PDT on albumin (BCA (bicinchoninic acid) protein assay). A spin gel filtration column was used to remove the toluidine blue pigment from the specimens. Although gel filtration reduced the amount of albumin by ~30%, application of a-PDT for 1 min or longer significantly reduced the amount of albumin (*p* < 0.05). (Albumin concentration was 2 mg/mL).

**Figure 4 antibiotics-10-00101-f004:**
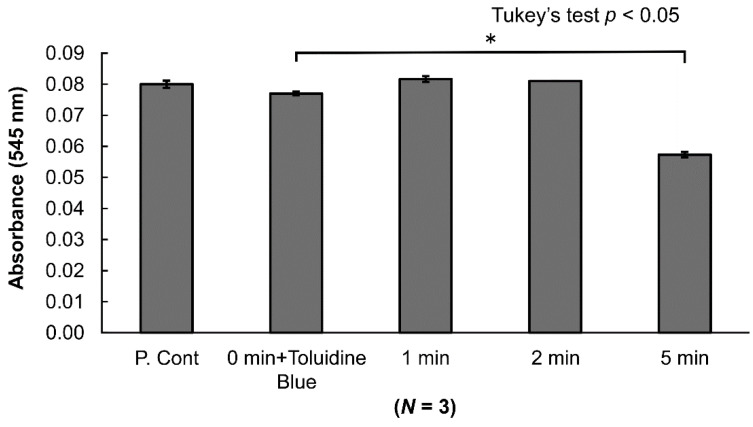
Effects of a-PDT on lipopolysaccharide (LPS) (chromogenic LAL (limulus amebocyte lysate) endotoxin assay). LPS measurement did not reveal any effects from a spin gel filtration column for pigment removal. Few changes in LPS were noted with a-PDT application of 1 or 2 min. However, LPS was decreased significantly (*p* < 0.05) with an application of 5 min. (LPS concentration was 1 EU/mL).

**Figure 5 antibiotics-10-00101-f005:**
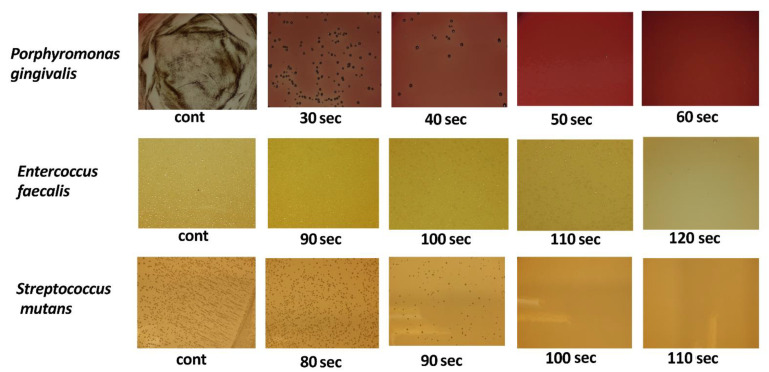
Antibacterial activity of a-PDT (bacterial culture testing). Bacteria were adjusted to 1 × 10^6^ CFU (colony forming unit)/mL; 0.01% concentration toluidine blue was added, and light was applied. Consequently, no additional colony formation was noted for *P. gingivalis* at 50 s, *E. faecalis* at 120 s, and *S. mutans* at 100 s.

**Figure 6 antibiotics-10-00101-f006:**
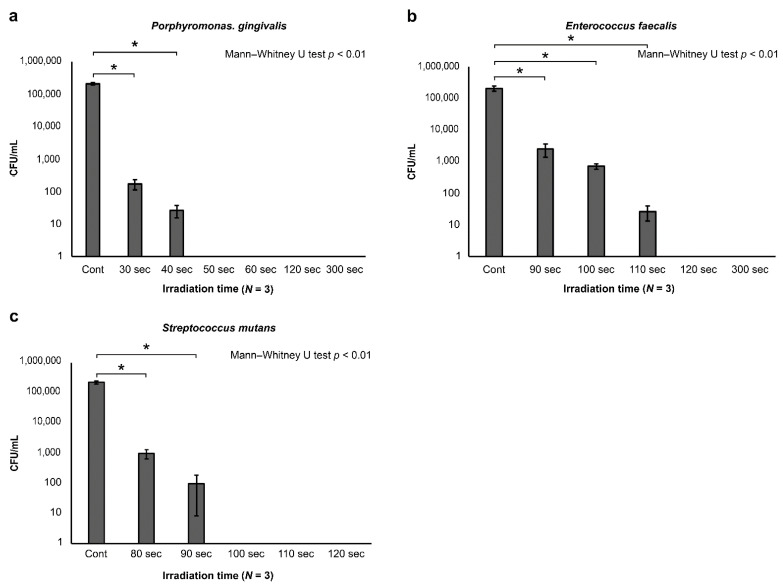
Bactericidal effects of bactericidal on each bacterial species (bacteria counts). After light application, specimens were cultured for seven days, after which colonies were counted. The results revealed that (**a**) *P. gingivalis* significantly decreased with 30 s of application and became undetectable with 50 s of application. Similarly, (**b**) *E. faecalis* was significantly decreased with 90 s of application and became undetectable with 120 s of application, while (**c**) *S. mutans* was significantly decreased with 80 s of application and became undetectable with 100 s of application.

## Data Availability

Data is contained within the article.

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
