# Peer review of "Effects of Antimicrobial Photodynamic Therapy on Organic Solution and Root Surface In Vitro"

_antibiotics, 2021, doi:10.3390/antibiotics10020101_

Round 1
Reviewer 1 Report
In the work, the authors investigated antimicrobial PDT effect in vitro on three bacteria including Porphyromonas gingivalis, Streptococcus mutans and Enterococcus faecalis that can potentially cause dental diseases. Overall, the design and the aim of the work are clear. And the data mostly supported the conclusion that the combination of FotoSan630 and toluidine blue could induce bactericidal efficacy as well as the decrease of organic substances. However, before publications, I suggested several revisions below.
1 The writing needs substantial improvement. A brief introduction of this project and its advantages are suggested to be added at the end of introduction section. The language needs polishing. Correct some typos. (e.g. bleu, Antrimicrobia and many others)
2 Please highlight the sediments in two SEM images for better visualization.
3 In Figure 5, it seems that it has the most bacteria for Enterococcus Faecalis at 100 sec. Please explain.
Author Response
"Please see the attachment."

Reviewer 2 Report
This is nice well-written paper, it may be accepted in present form.
Author Response
Dear Reviewer 2,
Thank you for your careful peer review.
We checked the spelling of the paper and corrected it.
Thank you very much.
Reviewer 3 Report
This study investigated the properties of PDT on bacteria and organic contents. It is of interest to better understand how works PDT. There are only minor comments:
Introduction is well written.
l52: planing
Results:
- the analysis based on SEM is only descriptive and lack quantification. It is quite impossible to conclude anything based only on these 2 images.
- please explain the rationale behind the evaluation of the albumin content and response to PDT
How do you explain the time related effect. An application during 5 minutes seems not applicable in clinic. Can you develop such point in the discussion section and interpret results according to existing pdt protocols.
Author Response
"Please see the attachment."

Reviewer 4 Report
The present paper is a very good paper.
I am glad to inform that I enjoy very much the reading.
Author Response
Reviewer 4,
Thank you for your careful peer review.
Thank you very much.